# Researching New Therapeutic Approaches for Abdominal Visceral Pain Treatment: Preclinical Effects of an Assembled System of Molecules of Vegetal Origin

**DOI:** 10.3390/nu12010022

**Published:** 2019-12-20

**Authors:** Carmen Parisio, Elena Lucarini, Laura Micheli, Alessandra Toti, Lorenzo Di Cesare Mannelli, Giulia Antonini, Elena Panizzi, Anna Maidecchi, Emiliano Giovagnoni, Jacopo Lucci, Carla Ghelardini

**Affiliations:** 1Department of Neuroscience, Psychology, Drug Research and Child Health-Neurofarba-Pharmacology and Toxicology Section, University of Florence, Viale Pieraccini 6, 50139 Florence, Italy; carmen.parisio@unifi.it (C.P.); elena.lucarini@unifi.it (E.L.); laura.micheli@unifi.it (L.M.); alessandra.toti91@gmail.com (A.T.); carla.ghelardini@unifi.it (C.G.); 2Aboca SpA Società Agricola, Innovation & Medical Science Division, Loc. Aboca 20, 52037 Sansepolcro (AR), Italy; GAntonini@aboca.it (G.A.); EPanizzi@naturalbiomedicine.it (E.P.); AMaidecchi@aboca.it (A.M.); EGiovagnoni@aboca.it (E.G.); JLucci@naturalbiomedicine.it (J.L.); 3Natural Bio-Medicine SpA, Loc. Aboca 20, 52037 Sansepolcro (AR), Italy

**Keywords:** visceral pain, colitis, IBDs, IBS, DNBS, Colilen IBS^®^, frankincense, aloe, chamomile, resins, polysaccharides, flavonoids, fractionation

## Abstract

Abdominal pain is a frequent symptom of irritable bowel syndrome (IBS) and inflammatory bowel diseases (IBDs). Although the knowledge of these pathologies is progressing, new therapeutic strategies continue to be investigated. In the present study, the effect of a system of molecules of natural origin (a medical device according to EU Directive 93/42/EC, engineered starting from *Boswellia serrata* resins, *Aloe vera* polysaccharides and *Matricaria chamomilla* and *Melissa officinalis* polyphenols) was evaluated against the intestinal damage and visceral pain development in DNBS-induced colitis model in rats. The system (250 and 500 mg kg^−1^) was orally administered once daily, starting three days before the injection of 2,4-dinitrobenzenesulfonic acid (DNBS) and for 14 days thereafter. The viscero-motor response (VMR) to colon-rectal balloon distension (CRD) was used as measure of visceral sensitivity. The product significantly reduced the VMR of DNBS-treated animals. Its effect on pain threshold was better than dexamethasone and mesalazine, and not lower than amitriptyline and otilonium bromide. At microscopic and macroscopic level, the tested system was more effective in protecting the intestinal mucosa than dexamethasone and mesalazine, promoting the healing of tissue lesions. Therefore, we suggest that the described system of molecules of natural origin may represent a therapeutic option to manage painful bowel diseases.

## 1. Introduction

Visceral pain represents a grievous form of pain that affects about 25% of the world population; abdominal pain is the most common gastrointestinal problem, encountered in a high percentage of adolescents, above all in women [1]. It can be the result of prolonged inflammatory processes, such as in inflammatory bowel diseases (IBDs), but in many patients, negative diagnostic test results lead to the diagnosis of irritable bowel syndrome (IBS) [2]. The two common forms of IBDs, ulcerative colitis (UC) and Crohn’s disease (CD), affect about 3.6 million of people in the United States and Europe, with chronic recurrent ulceration of the bowels resulting in an inflamed gut and the breakdown of the intestinal barrier function [3,4]. IBD causes significant symptoms such as diarrhea or constipation, nausea, cramping, abdominal pain, rectal bleeding, tiredness, weight loss and anxiety [5]. Some patients can go through periods during which the symptoms are particularly troublesome, followed by periods with very mild or absent symptoms [4]. Abdominal pain is a common manifestation in IBD, due to changes that begin with hypersensitivity of the primary sensory neurons, which innervate the gastrointestinal tract, and afterwards reach the central nervous system (CNS) [6]. Inflammation does not seem to completely explain altered perception of pain in patients affect by IBD, in fact 20%–50% of patients complain about abdominal pain in the clinical remission stage [6,7]. The poor correlations between reported abdominal pain intensity and IBD activity indices reinforce the complex nature of this type of pain and connect it to IBS [8,9]. IBS is a functional bowel disorder characterized by the presence of chronic/recurrent abdominal pain or discomfort, with altered bowel habits and consequent anomalies in stool frequency and form [10]. Patients may be categorized as diarrheal-predominant, constipation-predominant, or as having both [11]. Frequently, it is the result of a previous intestinal damage caused by infections or prolonged inflammatory processes [12]. Although IBD and IBS are different pathologies, there is a clinical overlap between these, with IBS-like symptoms reported in patients before the diagnosis of IBD or in remission stage. Therefore, the therapeutic approach to relieve pain in IBD and IBS is often the same [13]. In this context, finding valid therapies that can alleviate chronic visceral pain represents an important medical need to improve patients’ quality of life. The current therapy includes psychotherapy-behavioural, nutritional approaches and symptomatic treatments, such as stool bulking agents or laxatives, antidiarrheals (loperamide, colestyramine), antispasmodics (otilonium bromide), aminosalicylates (mesalazine or 5-ASA), corticosteroids (dexamethasone), tricyclic antidepressants (amitriptyline), opioids, and non-steroidal anti-inflammatory drugs (NSAIDs) [6,14,15]. Unfortunately, the current therapeutics to treat visceral pain offer little benefit for abdominal symptoms and can produce undesired and serious side effects [16]. In this context, the pain management remains unsatisfactory, so there is a growing demand for the development of effective treatments. Considering that visceral pain has been related to inflammatory processes [17], as well as to alterations in the enteric barrier and immune response [18], a product able to protect the intestinal mucosa by indirectly controlling inflammation and at the same time modulating immune response could be the answer. For this purpose, different classes of natural molecular complexes are receiving a lot of attention for their properties: the main ones are resins, polysaccharides, and polyphenols [19]. We selected a system of molecules capable of interacting with the intestinal mucosal surface forming a protective film, whose properties are engineered in order to achieve a beneficial action on swelling, abdominal distension, and pain by pooling frankincense (*Boswellia serrata*) resins, aloe (*Aloe vera* L.) polysaccharides, as well as chamomile (*Matricaria chamomilla* L.) and melissa (*Melissa officinalis* L.) polyphenols.

Aim of the present study was to evaluate the protective effect of the described system of molecules on the intestinal damage and visceral pain development in the rat model of colitis induced by DNBS. The effect of the system of molecules was compared to that obtained by reference drugs. The intrarectal injection of 2,4-dinitrobenzenesulfonic acid (DNBS) is able to induce colitis in rats with symptoms analogous to those observed in patients affected by intestinal inflammatory pathologies [20,21].

## 2. Material and Methods

### 2.1. System of Molecules Composition

The tested system of molecules (system) was produced in the form of 587 mg capsules containing, natural molecular complex (IT n° 102012902020829, EP2812013B1) composed of frankincense (*Boswellia serrata*) resinoid fraction (18.23% titrated in resins ≥ 60%), aloe (*Aloe vera* L.) polysaccharidic fraction (7.06% titrated in polysaccharides ≥ 35%), chamomile (*Matricaria chamomilla* L.) polysaccharidic fraction (3.52%, titrated in similar apigeninosimum flavonoids ≥ 0.2%), and melissa (*Melissa officinalis* L.) polysaccharidic fraction (31.60%, titrated in rosmarinic acid ≥ 1%). It also contains the essential oil of sweet fennel (1.43%), cumin powder (10%), and microcrystalline cellulose (9.93%). The remaining weight is represented by capsule weight.

### 2.2. Animals

Were used male Sprague-Dawley rats (arrived from Envigo, Varese, Italy) weighing around 200–250 g at the beginning of the experiments. Animals were housed in CeSAL (Centro Stabulazione Animali da Laboratorio, University of Florence) and they were used for the experimental procedures after one week. In each cage (size 26 × 41 cm), four animals were housed. Rats were fed with a standard laboratory diet and tap water ad libitum; they were kept at 23 ± 1 °C with a 12 h light/dark cycle (starting at 7 a.m.). The ethical policy of the University of Florence complies with the Guide for the Care and Use of Laboratory Animals of the US National Institutes of Health (NIH Publication No. 85-23, revised 1996; University of Florence assurance number: A5278-01). Formal approval to conduct the described experiments was obtained from the Animal Subjects Review Board of the University of Florence. All experiments were carried out according to the Directive 2010/63/EU of the European parliament and the European Union council (22 September 2010) on the protection of animals used for scientific purposes. Experiments involving animals have been reported according to ARRIVE guidelines [22]. All efforts were made to reduce the number of animals used and minimize their suffering.

### 2.3. Induction of Colitis

Colitis was induced in rats in conformity with the method described by Fornai et al. [23]. During a brief period of anesthesia with isoflurane (2%), 30 mg of 2,4-dinitrobenzenesulfonic acid (DNBS; Sigma-Aldrich, Milan, Italy) dissolved in 0.25 mL of 50% ethanol was intrarectally injected using a polyethylene PE-60 catheter inserted 8 cm proximal to the anus. Further, 0.25 mL of saline solution was injected in control rats.

### 2.4. Treatments

The tested system of molecules of natural origin (250 and 500 mg kg^−1^; Aboca S.P.A, Italy. LOT 17A1896) was solubilized in 1% CMC and orally administered once daily in the animals. Treatment started 3 days before DNBS injection (pre-treatment) and continued for 14 days after colitis induction. Control animals were treated with vehicle. Dexamethasone (1 mg kg^−1^; Carbosynth, Compton, United Kingdom), amitriptyline (15 mg kg^−1^; Sigma-Aldrich, Milan, Italy) and otilonium bromide (20 mg kg^−1^; Carbosynth, Compton, United Kingdom) were solubilized in 1% CMC and orally administered once daily for 14 days after DNBS injection. Mesalazine (50 mg kg^−1^: Asacol^®^, rectal suspension) was solubilized in saline solution and intrarectally administered once daily for 14 days after the colitis induction.

### 2.5. Assessment of Visceral Sensitivity

The viscero-motor response (VMR) to colorectal balloon distension (CRD) were used as objective measure of visceral sensitivity. In animals under deep anaesthesia, two EMG electrodes were sewn into the external oblique abdominal muscle and exteriorised dorsally [24]. VMR assessment were carried out under light anaesthesia (isoflurane 2%). A lubricated latex balloon (length: 4.5 cm), assembled to an embolectomy catheter and connected to a syringe filled with water was used to perform colon-rectal distension. A syringe was used to fill the balloon placed into the colon with various volumes of water (0.5, 1, 2, 3 mL). The electrodes were connected to a data acquisition system and the corresponding EMG signal consequent to colon-rectal stimulation were recorded, amplified, and filtered (Animal Bio Amp, ADInstruments, Colorado Springs, CO, USA), digitised (PowerLab 4/35, ADIinstruments, Colorado Springs, CO, USA), analysed and quantified using LabChart 8 (ADInstruments, Colorado Springs, CO, USA). To quantify the magnitude of the viscero-motor response at each distension volume, from the area under the curve (AUC) during the balloon distension (30 s) was subtracted the AUC immediately before the distension (30 s), and responses were expressed as percentage increase from the baseline. The time elapsed between two consecutive distension was 5 min. The measurements were carried out 3, 7, and 14 days after DNBS injection.

### 2.6. Macroscopic and Microscopic Analysis of Tissue Damage

On days 3, 7, and 14, the animals were sacrificed and the colon-rectal portion of the intestine was removed and processed for both macroscopic and microscopic analyses. A macroscopic damage score (MDS) was assigned to each animals in conformity to the criteria reported by Antonioli et al. [25]: presence and extension of hyperaemia and macroscopic mucosal damage (0–5); presence of adhesions between other intra-abdominal organs and colon (0–2); consistency of faeces (indirect marker of diarrhoea; 0–2); thickening of colonic wall (mm).

The intestine samples were then fixed in formalin at 4% for 24 h, dehydrated in alcohol, included in paraffin and finally cut into 5 μm sections. Histological evaluations were performed with an optical microscope on cross sections of haematoxylin and eosin stained colon, paying particular attention to the structure of the mucosa, the presence of cellular infiltrate, and the wall thickening. Micrographs to be analysed were taken using Nikon Olympus BX40 and a 400X objective equipped with NIS F3.00 Imaging Software^®^.

### 2.7. Statistical Analysis

All the experimental procedures were performed by researchers blind to the treatment. Behavioural measurements were performed on 6 animals for each group carried out in 2 different experimental sets. Results were expressed as mean ± S.E.M. The analysis of variance of behavioural data was performed by one-way ANOVA and a Bonferroni’s significant difference procedure was used as post-hoc comparison. *p*-values of less than 0.05 or 0.01 were considered significant. Data were analyzed using the “Origin 9” software (OriginLab, Northampton, MA, USA).

## 3. Results

### 3.1. Effect of Two Different doses of System and Reference Drugs on Visceral Hypersensitivity

Figure 1 shows the effect of repeated administration of two different doses of the system of molecules (250 and 500 mg kg^−1^ p.o.) on visceral hypersensitivity induced by intrarectal injection of 2,4-dinitrobenzenesulfonic acid (DNBS, 30 mg dissolved in 0.25 mL EtOH 50%) in the rats. A lubricated latex balloon (length: 4 cm) was inserted into the colon and positioned 8 cm from the anus. The magnitude of the abdominal contraction to each distension volume (0.5, 1, 2, 3 mL) of the balloon was recorded and used as an objective measure of visceral sensitivity in the animals. The product was administered daily starting from 3 days before DNBS injection up to 14 days after induction of the damage. Saline solution was intra-rectally injected into control animals. The assessment of visceral sensitivity was performed in animals 7 (Figure 1a) and 14 (Figure 1b) days after DNBS injection, measuring the abdominal viscero-motor response (VMR) to colon-rectal distension (CRD). The visceral sensitivity resulted significantly higher in DNBS-treated animals in comparison to controls 7 and 14 days after DNBS injection.

In animals treated with DNBS + system 250 mg kg^−1^ the abdominal viscero-motor response to colon-rectal distension was significantly reduced with 3 mL balloon inflation compared to animals treated with DNBS + vehicle, both 7 (Figure 1a) and 14 (Figure 1b) days after the induction of the damage.

At 3 (Appendix A) and 7 (Figure 1a) days from the induction of the damage were not observed significant differences between the visceral sensitivity of the animals treated with DNBS + system 500 mg kg^−1^ and that of animals treated with DNBS + vehicle. On the contrary, after 14 days from the induction of colitis, the abdominal viscero-motor response to colon-rectal distension was considerably reduced in animals treated with DNBS + system 500 mg kg^−1^, compared to animals treated with DNBS + vehicle, with all balloon distension volumes, in a statistically significant manner.

Figure 2 shows the effect of repeated treatment with dexamethasone (1 mg kg^−1^ p.o.), amitriptyline (15 mg kg^−1^ p.o.), mesalazine (50 mg kg^−1^ i.r.) and otilonium bromide (20 mg kg^−1^ p.o.) on visceral hypersensitivity induced by intra-rectal injection of DNBS (30 mg dissolved in 0.25 mL EtOH 50%) in the rats. Each compound was administered daily from the day of DNBS injection, up to 14 days after the induction of the damage. Visceral sensitivity was assessed in animals 7 (Figure 2a) and 14 (Figure 2b) days after DNBS injection. Repeated administration of dexamethasone did not show any effect at 7 days from the induction of the damage (Figure 2a), while at 14 days it significantly reduced the visceral hypersensitivity in the animals (Figure 2b). Mesalazine was not able to significantly reduce visceral hypersensitivity, although at 14 days a decrease in abdominal viscero-motor response was observed in animals (Figure 2b). On the other hand, repeated administration of amitriptyline and otilonium bromide were effective in reducing visceral hypersensitivity in animals both at 7 and 14 days after induction of damage (Figure 2).

### 3.2. Effect of System Injection on Colon Damage

Figure 3 and Figure 4 show the effect of repeated treatment with system 500 mg kg^−1^ on tissue damage induced by DNBS at the colon-rectal level. The animals were sacrificed 3, 7, and 14 days after DNBS administration and the colon was harvested and processed for both macroscopic (Figure 3) and microscopic (Figure 4) histological analysis.

The macroscopic damage score (MDS) was used to quantify the tissue damage degree. The MDS was evaluated on the newly explanted tissue assigning to each animal a score based on: presence and extension of hyperaemia and macroscopic mucosal damage (0–5); presence of adhesions between other intra-abdominal organs and colon (0–2); consistency of faeces (indirect marker of diarrhoea; 0–2); thickening of colonic wall (mm). In animals treated with DNBS + vehicle the DMS was significantly higher about 9, 8, and 6 times than control animals score at 3, 7 and 14 days, respectively, from the induction of the damage. In animals treated with DNBS + system 500 mg kg^−1^ a statistically significant reduction of the MDS was observed compared to that of animals treated with DNBS + vehicle at 7 and 14 days after injury induction of approximately 5 and 4 times, respectively (Figure 3). Repeated administration of system 250 mg kg^−1^ at 14 days after the induction of the injury resulted in a reduction (although not significant) of DMS (Appendix A).

Histological evaluation of the damage at the microscopic level was subsequently performed on sections of intestine stained with Hematoxylin and Eosin, paying particular attention to the structure of the mucosa, the presence of cellular infiltrate, and the eventual thickening of the muscular tissue. The microscopic observation of the colon of the control animals did not reveal significant alterations at the tissue level, and this allowed to exclude that the lesions observed in the other experimental groups were consequent to the method used (Figure 4).

Three days after the damage-induction, the colon of the animals treated with DNBS + vehicle appeared thickened with extensive ulcerations characterized by coagulation necrosis, as evidenced by the high number of neutrophils and mononuclear cells present around the lesions (black arrow and white cross, Figure 4). This pathological situation has emerged both in the transverse and longitudinal section of the mucosa. The lesions observed in animals treated with DNBS + system 500 mg kg^−1^ 3 days after the induction of the damage were similar to those examined in animals treated with DNBS + vehicle.

On day 7, the colon of animals treated with DNBS + vehicle showed extensive epithelial damage due to the formation of ulcers with loss of mucosa (white cross, Figure 4), transmural immune cell infiltration (predominantly neutrophils and lymphocytes, black arrow Figure 4 10×), crypts abscesses (black arrow; Figure 4 40×), endothelial cell hyperplasia with elongation (white arrow, Figure 4) and deformation (green arrow, Figure 4) of crypts and goblet cells hypertrophy, index of mucosal hypersecretion (yellow arrow, Figure 4). The colon of the animals treated with DNBS + system 500 mg kg^−1^ 7 days after the induction of the damage did not show ulcerations. The inflammatory infiltrate was reduced and limited to the mucous and submucosal portion, compared to animals treated with DNBS + vehicle. Endothelium hyperplasia (white arrow, Figure 4) and diffuse goblet cells hypertrophy (yellow arrow, Figure 4), which shows a high increase in mucosal secretion, has been observed. The crypts appeared to vary in size (green arrow, Figure 4) but the presence of abscesses inside them was not detected.

The colon of DNBS treated animals on day 14 appeared thick and hyperaemic with inflammatory infiltration both at mucosal and transmural level (black arrow, Figure 4). A large and widespread increase in neutrophils both in the epithelial cells and in the lumen of the crypts was detected (black arrow, Figure 4, 40×). The crypts showed an irregular structure, with a variable diameter (hyperplasia of the crypts, green arrow, Figure 4) and a small number of goblet cells (white arrow, Figure 4). Furthermore, on the epithelial surface, the presence of eschar and granulocytic-fibrous tissue was observed, probably resulting from the cicatrisation of previous ulcers (white cross, Figure 4). In the animals treated with the system of molecules 500 mg kg^−1^ the inflammatory infiltrate was reduced and almost exclusively limited to the submucosa, with an overall reduction of neutrophils. The structure and dimensions of the crypts were regular and comparable to those of control animals. Goblet and epithelial cells hyperplasia, as well as thickening of the intestinal wall and increase in mucous secretion was not observed. In addition, the tissue was found to be free of eschar or fibrotic cicatricial portions.

At 14 days from the induction of the damage, in the animals treated with DNBS + system 250 mg kg^−1^ the presence of inflammatory infiltrate was evident both in the mucosa and in the underlying structures (black arrow, Appendix A), with a high number of neutrophils. Crypts structure was generally regular with bifurcations (green arrow, Appendix A). 

### 3.3. Effect of Reference Drugs Administration on Colon Damage

Figure 5 and Figure 6 show the effect of repeated injection of dexamethasone (1 mg kg^−1^ p.o.), amitriptyline (15 mg kg^−1^ p.o.), mesalazine (50 mg kg^−1^ i.r.) and otilonium bromide (20 mg kg^−1^ p.o.) on tissue damage induced by DNBS at the colon-rectal level. The extent of macroscopic (Figure 5) and microscopic (Figure 6) damage on the explanted tissue was assessed (as described above) 14 days after DNBS injection.

Both repeated treatment with dexamethasone and mesalazine was able to significantly reduce the macroscopic *score* in animals treated with DNBS about 2 and 3 times, respectively, while treatment with amitriptyline and otilonium bromide did not show a protective effect on tissue damage (Figure 5).

At the microscopic evaluation, the colon of animals treated with DNBS + amitriptyline and DNBS + otilonium bromide showed the same type of alterations observed in animals treated with DNBS + vehicle (Figure 6): significant structural alterations in the crypts (white and green arrow, Figure 6) and eschar on the intestinal surface (white cross, Figure 6) were observed, although less extensive than those found in animals administered with DNBS + vehicle. The colon of the animals administered with DNBS + mesalazine did not show significant alterations in the structure of the mucosa, although some crypts showed hyperplasia (white arrow, Figure 6) and increased secretion of mucus. The presence of inflammatory infiltrate was limited to mucosal and submucosal (black arrow, Figure 6).

## 4. Discussion

In the present study, we reported the efficacy of a system of molecules of natural origin, a medical device made of natural substances according to EU Directive 93/42/EC, against the development of intestinal damage and visceral pain in the rat model of DNBS-induced colitis. The system is able to effectively protect the intestinal mucosa, promoting its healing processes, above all, it is able to significantly reduce the visceral hypersensitivity in DNBS-injected rats.

DNBS was frequently used as model of colitis [26,27]; recently we describe the possibility to use the same model for studying the post-inflammatory phase highlighting pain chronicization [28]. DNBS induces a local inflammation peaking between 3 and 7 days after injection, with release of cytokines such as interferon-γ (IFN-γ), tumor necrosis factor-α (TNF-α) and interleukin-12 (IL-12) [29]. During this phase, the appearance of a series of IBS-like symptoms is observed: visceral hypersensitivity, motility disorder and alterations of the permeability or the secretion that also persist in the animals during the phase of recovery from the colitis [30]. Moreover, DNBS also causes IBD-like symptoms such as intestinal mucosal and transmural damage, intestinal motility dysfunction, release of pro-inflammatory cytokines, and dysbiosis [31].

Despite the large number of sufferers, the underlying pathophysiology of visceral hypersensitivity in these common disorders remains obscure. Different mechanisms have been proposed to contribute to the initiation, exacerbation and persistence of visceral pain. Among the pathophysiological mechanisms involved, psychosocial factors, alterations of gastrointestinal motility and visceral sensitivity play an important role [32]. Other studies have also identified factors such as neuroplasticity, alterations in serotonin metabolism, gastrointestinal infections, microbiota alterations and diet [33,34]. The importance of both epithelial barrier function and innate immunity seem to perform a key role in the development of intestinal inflammation [35]. The intestinal barrier is a functional unit divided into different structures, whose integrity is necessary for the maintenance of normal intestinal permeability [36]. The modification of this complex balance can cause the passage of the luminal content towards the underlying tissues, and therefore into the blood circulation, with alteration of the normal physiological functions of the barrier [36,37]. These changes cause the activation of the immune response with consequent induction of an inflammatory state [38,39]. In some pathologies, as in IBDs, IBS and celiac disease, the alteration of intestinal permeability could represent the *primum movens* of the pathology [40]. The production of cytokines, including IFN-γ and TNF-α, secondary to the inflammatory process, causes the increase in intestinal permeability, with development of chronic visceral hypersensitivity [41].

In this context, therefore, new therapeutic strategies focused at the same time on the recovery of the physiological function of the intestinal barrier, on the indirect control of inflammatory processes and modulation of the immune response, can offer an innovative approach for the improvement of the clinical picture of these chronic diseases.

The system is designed in order to interact with the surface of the intestinal mucosa forming a protective film, with a beneficial action on swelling, pain, and abdominal distension, by pooling materials such as frankincense resins, *Aloe vera* polysaccharides, as well as chamomile and melissa polyphenols.

The oleo-gum resin from *Boswellia serrata* (*Burseraceae*), or Indian frankincense, contains chemical constituents such as alkaloids, terpenoids, tannins, phenols, saponins and triterpenes. These compounds have been largely used in traditional Indian medicine for their many beneficial properties such as anti-inflammatory and antiarthritic effects [42], but their efficacy in mouse models of chemically induced colitis is debated [43]. Specific evidence of the frankincense protection properties on membrane integrity was found by Catanzaro et al. [16] on Caco-2 human intestinal cells, subjected to inflammation mediators (INF-γ or TNF-α) or to hydrogen peroxide (source of free radicals). The results show that frankincense significantly preserves the tight junctions from the damage caused by free radicals and inflammatory mediators, reducing their levels. This protective action is also reflected in the paracellular permeability and cellular integrity and is not completely recapitulated by single components of the mixture of molecules composing the extract [16]. In vivo experiments, performed in different colitis models in rats, showed the antioxidant properties of the *Boswellia serrata* extracts, with intestinal anti-inflammatory effect [44], mainly attributed to leukotrienes (LT) inhibition, which play a role in the initiation and continuance of inflammation. Their inhibition can avoid the oxidation of lipids and release of inflammatory cytokines [45]. Hartmann et al. [46] also demonstrate a significant reduction in lipid peroxidation, nitric oxide and inducible nitroxidase (iNOS) in frankincense-treated animals.

Many studies have proven the medicinal effects of *Aloe vera* extracts, exercised through polysaccharides, which are the main functional components, with bio-adhesive activities [47]. It is also widely known that *Aloe vera* exerts its protective properties of restoring and maintaining the intestinal mucosa integrity alone, or in combination with other therapies [48,49]: Azar et al. [48] show the effect of a mixture of chamomile and *Aloe vera* on colitis model in rats, with protective and antioxidant action.

Chamomile and Melissa extracts have many effects including mild sedative and carminative, spasmolytic, antibacterial, antiviral, and antioxidative effects [50]. The polyphenols present in chamomile and melissa are flavonoids containing quercetin, luteolin, apigenine-like and other substances with similar chemical characteristics that allow the compound to become a preferential substrate for free radicals, with radical scavenger action [51]. Albrecht et al. [52] showed the protective action of a formulation containing myrrh, coffee charcoal and chamomile flower extract on patients with symptoms of acute diarrhoea due to IBS. The combination was safe, well tolerated by patients, and demonstrated effects comparable to conventional pharmacological therapies [52].

Considering the impact of chronic pain on patient’s quality of life, there is an urgent need of developing new effective therapeutic strategies. The current therapeutic approach is ineffective and with side effects. For this reason, the use of natural products could represent a valid alternative.

Nevertheless, despite the many elegant and thorough characterisations of the biological effects elicited by mixtures of compounds of natural origin, a comprehensive and mechanistic depiction of the molecular underpinnings behind their efficacy is usually not enough explored. From a purely technological standpoint, in fact, current scientific knowledge does not even allow researchers to describe the pharmacodynamic and kinetic properties of a mixture of compounds in terms of the sum of the properties of its individual components. This would, anyway, represent a conceptual approximation of the actual subject matter [53,54], as such an approach would fail to consider any of the supramolecular phenomena, synergisms and antagonisms, that occur in the context of a mixture of compounds. Importantly, these phenomena are nowadays known to potentially alter the pharmacodynamic and kinetic profiles of the individual components in a mixture, thus generating profiles that are different to those that are observed when the same compounds are studied as isolated [55,56]. It becomes apparent that ad hoc designed experimental setups are needed in order to determine the peculiarities of mixtures [57].

The overall action that tested product exerts on the intestinal mucosa, is to protect it through the formation of a barrier and thus to reduce irritation, inflammation and pain. Although pre-treatment with the system does not prevent the development of DNBS-induced damage (3 days after treatment, damage is comparable to that observed in animals treated with DNBS + vehicle), this system of molecules is able to effectively protect the intestinal mucosa, promoting its healing processes. Compared to clinically-used compounds for abdominal pain relief in patients with IBDs or IBS, the effect of the system on pain threshold was better than that obtained with dexamethasone and mesalazine, and not lower than amitriptyline and otilonium bromide. At histological level, the system is more effective in protecting the intestinal mucosa than dexamethasone and mesalazine, the only reference compounds with protective characteristics. Even at the macroscopic level, the *score* obtained from the animals treated with the system is clearly lower than that of the animals treated with dexamethasone and mesalazine, suggesting a greater protective effect.

## 5. Conclusions

Our data confirm the ability of the system of molecules to protect the intestinal mucosa and to promote its healing from lesions, as demonstrated at histological level. The results show also the action that this product has on visceral pain, significantly reducing the onset of visceral hypersensitivity resulting from the damage. Therefore, we suggest that this product could represent an innovative therapeutic option to pain management in bowel diseases.

## 6. Patents

EP2812813B1.

## Figures and Tables

**Figure 1 nutrients-12-00022-f001:**
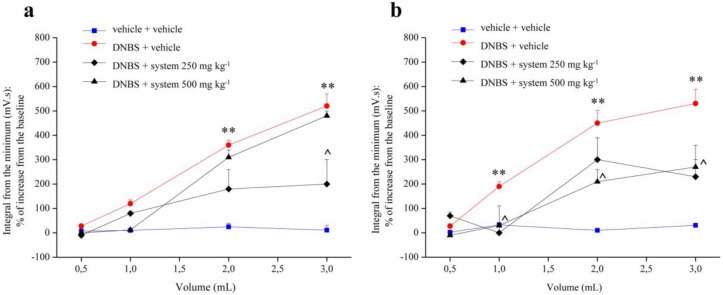
Effect of system administration on visceral hypersensitivity. Visceral hypersensitivity was provoked in rats by the intra-rectal injection of 2,4-dinitrobenzenesulfonic acid (DNBS, 30 mg in 0.25 mL EtOH 50%). Control animals were injected intra-rectally with saline solution. System (250 and 500 mg kg^−1^) was orally administered once daily in the animals: treatment started 3 days before DNBS injection and continued for 14 days after colitis induction. Tests were performed on day 7 (**a**) and 14 (**b**) after the damage induction, measuring the viscero-motor response to the colon rectal distension (0.5, 1, 2, 3 mL balloon inflation). Each value is the mean ± S.E.M. and represents the mean of 6 rat per group. ** *p* < 0.01 vs. vehicle + vehicle treated animals. ^ *p* < 0.05 vs. DNBS + vehicle treated animals.

**Figure 2 nutrients-12-00022-f002:**
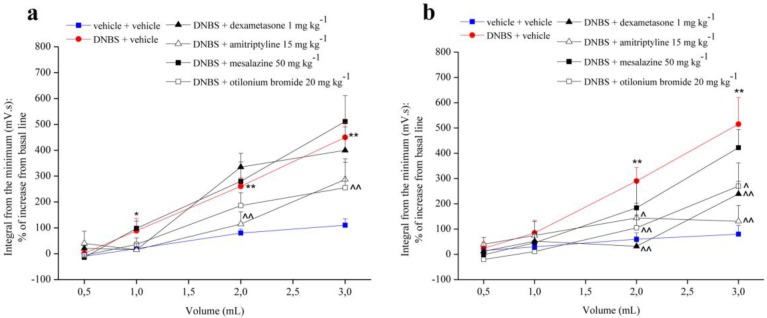
Effect of reference drugs administration on visceral hypersensitivity. Visceral hypersensitivity was provoked in rats by the intra-rectal injection of 2,4-dinitrobenzenesulfonic acid (DNBS, 30 mg in 0.25 mL EtOH 50%). Control animals were injected intra-rectally with saline solution. Dexamethasone (1 mg kg^−1^ p.o.), amitriptyline (15 mg kg^−1^ p.o.), otilonium bromide (20 mg kg^−1^ p.o.) and mesalazine (50 mg kg^−1^ i.r.) were administered once daily in the animals: treatment started the same day of DNBS injection and continued for 14 days after colitis induction. Tests were performed on day 7 (**a**) and 14 (**b**) after the damage induction, measuring the viscero-motor response to the colon rectal distension (0.5, 1, 2, 3 mL balloon inflation). Each value is the mean ± S.E.M. and represents the mean of 6 rat per group. * *p* < 0.05 and ** *p* < 0.01 vs. vehicle + vehicle treated animals. ^ *p* < 0.05 and ^^ *p* < 0.01 vs. DNBS + vehicle treated animals.

**Figure 3 nutrients-12-00022-f003:**
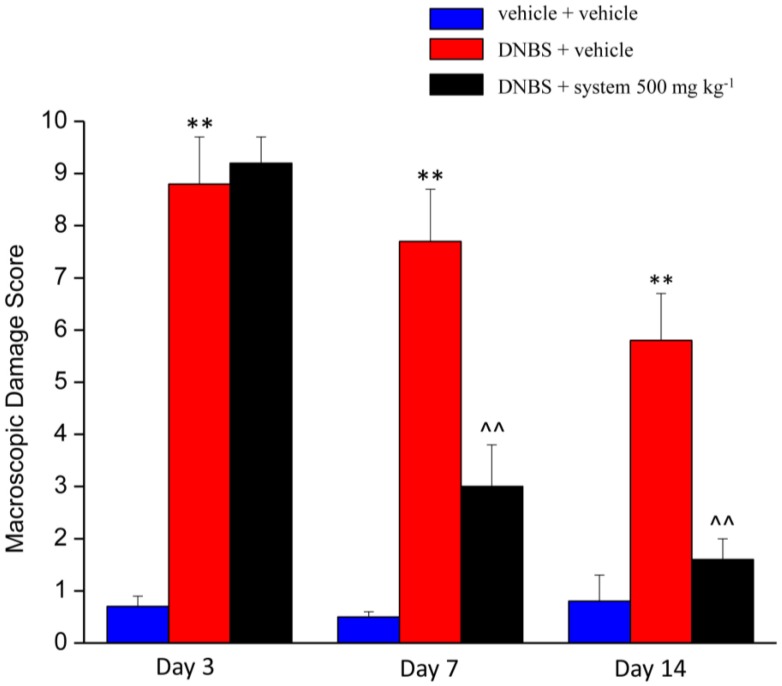
Effect of system administration on colon macroscopic damage. Visceral hypersensitivity was provoked in rats by the intra-rectal injection of 2,4-dinitrobenzenesulfonic acid (DNBS, 30 mg in 0.25 mL EtOH 50%). Control animals were injected intra-rectally with saline solution. System 500 mg kg^−1^ was orally administered once daily in the animals: treatment started 3 days before DNBS injection and continued for 14 days after colitis induction. Animals were sacrificed 3, 7 and 14 days after DNBS injection. A Macroscopic Damage Score (MDS) was assigned to each animals based on: presence and extension of hyperaemia and macroscopic mucosal damage (0–5); presence of adhesions between other intra-abdominal organs and colon (0–2); consistency of faeces (indirect marker of diarrhoea; 0–2); thickening of colonic wall (mm). Each value is the mean ± S.E.M. and represents the mean of 6 rat per group. ** *p* < 0.01 vs vehicle + vehicle treated animals. ^^ *p* < 0.01 vs. DNBS + vehicle treated animals.

**Figure 4 nutrients-12-00022-f004:**
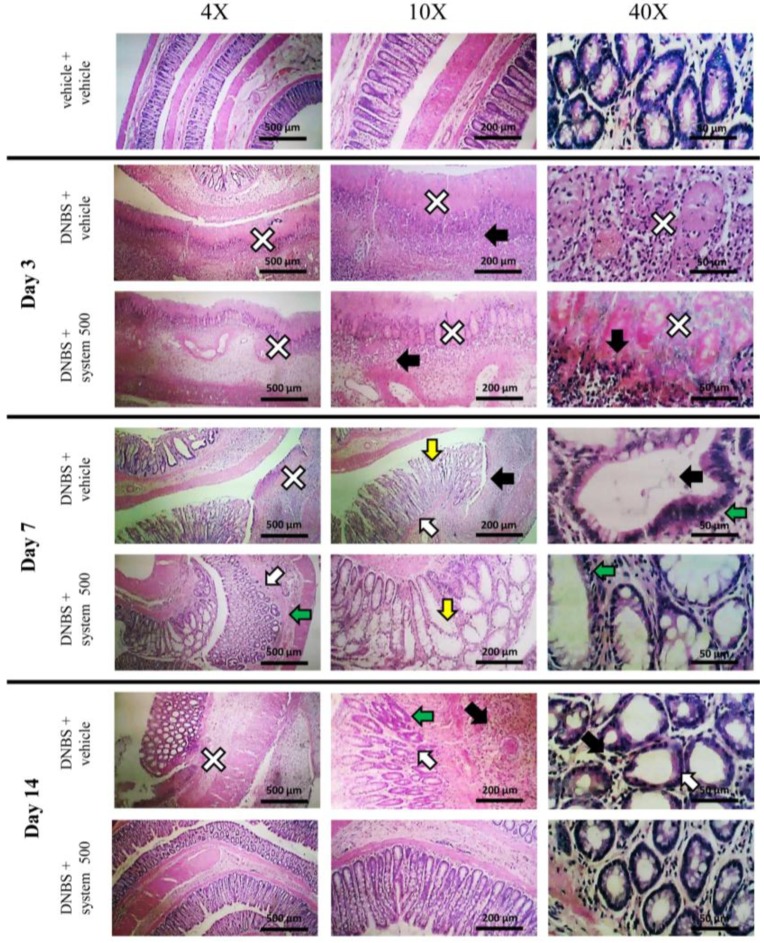
Effect of system administration on colon microscopic damage. Visceral hypersensitivity was provoked in rats by the intra-rectal injection of 2,4-dinitrobenzenesulfonic acid (DNBS, 30 mg in 0.25 mL EtOH 50%). Control animals were injected intra-rectally with saline solution. System 500 mg kg^−1^ was orally administered once daily in the animals: treatment started 3 days before DNBS injection and continued for 14 days after colitis induction. Animals were sacrificed 3, 7 and 14 days after DNBS injection. Microscopic evaluations were carried out by light microscopy on haematoxylin- and eosin- stained sections of colon (5 μm). In the Figure, it was highlighted the presence of: inflammatory mucosal cell infiltrate, submucosal and transmural, neutrophils between epithelial cells and neutrophils in crypt lumen (black arrow); granulocyte-fibrin eschar, loss of surface epithelium (white cross); goblet cell hyperplasia and mucus hypersecretion (yellow arrow); irregular crypts, variable crypts diameters, bifurcation and branched crypts (green arrow); epithelial hyperplasia, visible as crypt elongation, with goblet cell loss (white arrow).

**Figure 5 nutrients-12-00022-f005:**
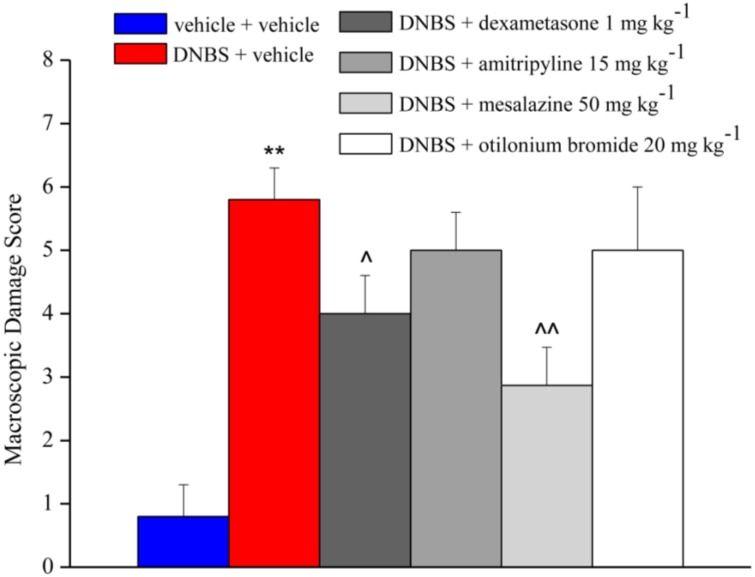
Effect of reference drugs administration on colon macroscopic damage. Visceral hypersensitivity was provoked in rats by the intra-rectal injection of 2,4-dinitrobenzenesulfonic acid (DNBS, 30 mg in 0.25 mL EtOH 50%). Control animals were injected intra-rectally with saline solution. Dexamethasone (1 mg kg^−1^ p.o.), amitriptyline (15 mg kg^−1^ p.o.), otilonium bromide (20 mg kg^−1^ p.o.) and mesalazine (50 mg kg^−1^ i.r.) were administered once daily in the animals: treatment started the same day of DNBS injection and continued for 14 days after colitis induction. On day 14 the animals were sacrificed and a Macroscopic Damage Score (MDS) was assigned to each animals based on: presence and extension of hyperaemia and macroscopic mucosal damage (0–5); presence of adhesions between other intra-abdominal organs and colon (0–2); consistency of faeces (indirect marker of diarrhoea; 0–2); thickening of colonic wall (mm). Each value is the mean ± S.E.M. and represents the mean of 6 rat per group. ** *p* < 0.01 vs. vehicle + vehicle treated animals. ^^ *p* < 0.01 and ^ *p* < 0.05 vs. DNBS + vehicle treated animals.

**Figure 6 nutrients-12-00022-f006:**
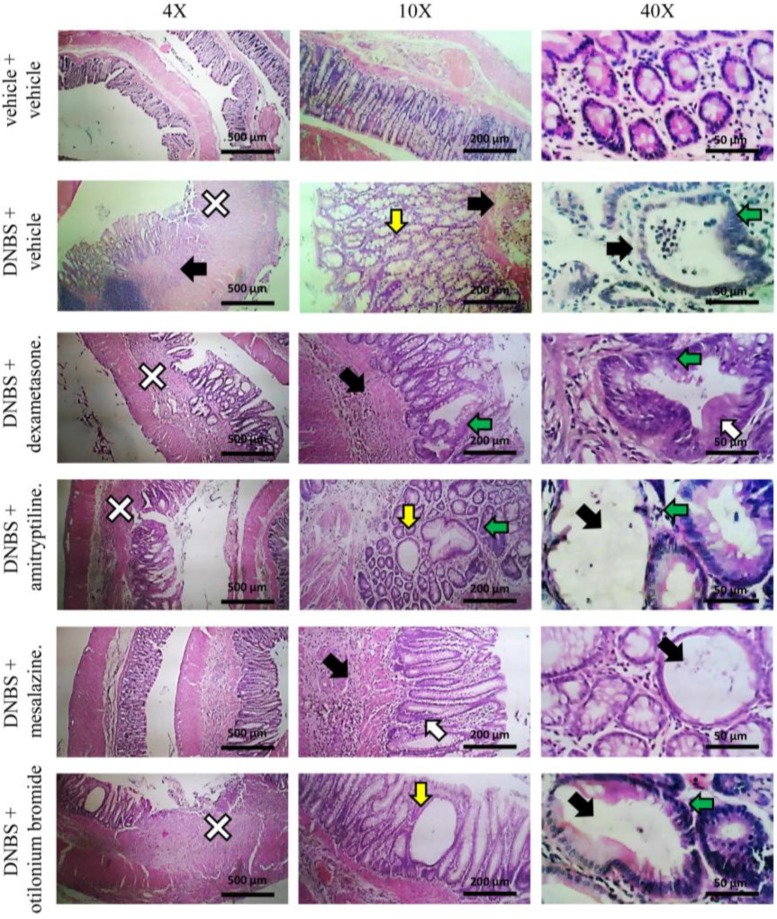
Effect of reference drugs administration on colon microscopic damage. Visceral hypersensitivity was provoked in rats by the intra-rectal injection of 2,4-dinitrobenzenesulfonic acid (DNBS, 30 mg in 0.25 mL EtOH 50%). Control animals were injected intra-rectally with saline solution. Dexamethasone (1 mg kg^−1^ p.o.), amitriptyline (15 mg kg^−1^ p.o.), otilonium bromide (20 mg kg^−1^ p.o.) and mesalazine (50 mg kg^−1^ i.r.) were administered once daily in the animals: treatment started the same day of DNBS injection and continued for 14 days after colitis induction. The animals were sacrificed on day 14. Microscopic evaluations were carried out by light microscopy on haematoxylin- and eosin- stained sections of colon (5 μm). In the Figure it was highlighted the presence of: inflammatory mucosal cell infiltrate, submucosal and transmural, neutrophils between epithelial cells and neutrophils in crypt lumen (black arrow); granulocyte-fibrin eschar, loss of surface epithelium (white cross); goblet cell hyperplasia and mucus hypersecretion (yellow arrow); irregular crypts, variable crypts diameters, bifurcation and branched crypts (green arrow); epithelial hyperplasia, visible as crypt elongation, with goblet cell loss (white arrow).

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
