# Peer review of "Researching New Therapeutic Approaches for Abdominal Visceral Pain Treatment: Preclinical Effects of an Assembled System of Molecules of Vegetal Origin"

_nutrients, 2019, doi:10.3390/nu12010022_

Round 1

Reviewer 1 Report

Major issues:

Figure 1. The VMR to CRD in control animals (vehicle + vehicle) does not change with the volume of the balloon. How do authors explain it? Even in control animals a response to increasing volume of balloon should be clearly visible. Otherwise it may be the case that the signal is not coming through electrodes to the amplifier or there are some other issues related to signal transduction in this setup. Furthermore, as showed on Fig. 2 VMR to CRD increases in control animals (approx. 100% increase) as the pressure increases.

Material and Methods: Chemical composition of Colilen IBS® should be mentioned in the material and methods section. Information provided by the Authors is too general and lacks specifics about the active compounds present in the tested preparation. This data is critical for the understanding of the mechanism of action of Colilen IBS®.

Conclusions: author claim that: “Our data confirm the ability of Colilen IBS® to protect the intestinal mucosa against irritating and inflammatory agents…” however several lines above in the discussion section author admit that “Although pre-treatment with Colilen IBS® does not prevent the development of DNBS-induced damage…”. Thus this overstatement has to be removed from the conclusions.

Minor issues:

Abstract: „Inflammatory Bowel Syndrome” should be corrected since the correct name id irritable bowel syndrome.

Abstract and discussion: “device” is not a proper word to use when describing a plant-derived product. Please amend.

Discussion: DNBS model should not be referred to as IBS model since it induces colonic inflammation which may or may not cause IBS. DNBS should be rather referred to as model of colitis.

Author Response

- Question 1. Figure 1. The VMR to CRD in control animals (vehicle + vehicle) does not change with the volume of the balloon. How do authors explain it? Even in control animals a response to increasing volume of balloon should be clearly visible. Otherwise it may be the case that the signal is not coming through electrodes to the amplifier or there are some other issues related to signal transduction in this setup. Furthermore, as showed on Fig. 2 VMR to CRD increases in control animals (approx. 100% increase) as the pressure increases.VMR to CRD in control animals (vehicle + vehicle) does not change with the volume of the balloon. How do authors explain it? Even in control animals a response to increasing volume of balloon should be clearly visible. Otherwise it may be the case that the signal is not coming through electrodes to the amplifier or there are some other issues related to signal transduction in this setup. Furthermore, as showed on Fig. 2 VMR to CRD increases in control animals (approx. 100% increase) as the pressure increases.

Response: We are grateful to reviewer for underlying this point. In our animal model, control rats (vehicle + vehicle) show a viscero-motor response to increasing volume of balloon that isn't always clearly visible (as shown in figure 1); in fact their response to the CRD is always slight, and it is significantly lower compared to animals treated with DNBS + vehicle. Furthermore, in figure 2 the increase of viscero-motor response to colon rectal balloon distension in control animals is minimal and it is not statistically significant respect to the responses obtained to smaller volumes. Therefore, considering also the biological variability of animals, we suggest that the recorded VMR really comes from the electrodes implanted in abdominal muscle and that it faithfully shows the response of the animals to CRD.viscero-motor response to increasing volume of balloon that isn't always clearly visible (as shown in figure 1); in fact their response to the CRD is always slight, and it is significantly lower compared to animals treated with DNBS + vehicle. Furthermore, in figure 2 the increase of viscero-motor response to colon rectal balloon distension in control animals is minimal and it is not statistically significant respect to the responses obtained to smaller volumes. Therefore, considering also the biological variability of animals, we suggest that the recorded VMR really comes from the electrodes implanted in abdominal muscle and that it faithfully shows the response of the animals to CRD.

- Question 2. Material and Methods: Chemical composition of Colilen IBS® should be mentioned in the material and methods section. Information provided by the Authors is too general and lacks specifics about the active compounds present in the tested preparation. This data is critical for the understanding of the mechanism of action of Colilen IBS®.

Response: We understand the doubts of the reviewer about the information provided on the composition of the Colilen. About this, we refer to an additional patent, different from the one present in the original version of the manuscript. The document is freely available online (e.g. https://patents.google.com/patent/EP2812013B1/en and attached as Annex III to this document) and includes part of the data that clarified the mechanism underlying product efficacy. As partially shown in the patent, the characterized formulation was designed through an extensive research work which conferred to the complex mixture of molecules characteristic features that underlie its efficacy. For example, bioadhesion and the ability to build a barrier against proinflammatory stimuli (thus resulting in an antinflammatory activity). It is crucial to consider that all experiments conducted refers to measurement of properties of the final product as a whole. Such properties are therefore to be defined as “emerging properties”, as they are typical of the final mixture and not of its single molecular components. From this aspect, also derives that properties of the single components investigated as isolated, cannot be ascribed to the mixture. The application of this combination of technical approaches, particularly necessary within the relevant legislative framework, factually replaces the concept of active pharmaceutical ingredient dissolved within a pool of excipients with the concept of system of molecules, designed in order to acquire desirable emerging properties. Having successfully experimentally demonstrated to the competent authorities(in the context of the relevant technical file) that product efficacy should be ascribed to emerging properties such as bioadhesion and barrier effect, we turned our attention to what would be the best possible way to deliver the relationship between composition and its resulting biological activity. In our interpretation, this appears to us to be very correctly also the spirit of the reviewer’s concern. Building on the know-how we accumulated over the R&D process, we concluded that it would be correct, and were comforted by the agreement expressed by regulatory authorities, to refer to the contribution of polysaccharides and resins to the adhesion and barrier properties of the product. It is of pivotal important, though, to consider that for instance the adhesive and barrier properties of the final product cannot be reconducted exclusively to the presence of these single fractions but to properties that emerge contemporarily in the context of the final formulation and are determined by the interactions between the different components. As a consequence of this we actually also centered the batch release policy of the product on reproducibility of these parameters, which are the ones underlying its efficacy. We sincerely hope that in the light of this explanation, the reviewer will be able to accept our revision of the manuscript, in which we add an additional patent and quantitative information with respect to the abundance of different extracts in the final product.Colilen. About this, we refer to an additional patent, different from the one present in the original version of the manuscript. The document is freely available online (e.g. https://patents.google.com/patent/EP2812013B1/en and attached as Annex III to this document) and includes part of the data that clarified the mechanism underlying product efficacy. As partially shown in the patent, the characterized formulation was designed through an extensive research work which conferred to the complex mixture of molecules characteristic features that underlie its efficacy. For example, bioadhesion and the ability to build a barrier against proinflammatory stimuli (thus resulting in an antinflammatory activity). It is crucial to consider that all experiments conducted refers to measurement of properties of the final product as a whole. Such properties are therefore to be defined as “emerging properties”, as they are typical of the final mixture and not of its single molecular components. From this aspect, also derives that properties of the single components investigated as isolated, cannot be ascribed to the mixture. The application of this combination of technical approaches, particularly necessary within the relevant legislative framework, factually replaces the concept of active pharmaceutical ingredient dissolved within a pool of excipients with the concept of system of molecules, designed in order to acquire desirable emerging properties. Having successfully experimentally demonstrated to the competent authorities(in the context of the relevant technical file) that product efficacy should be ascribed to emerging properties such as bioadhesion and barrier effect, we turned our attention to what would be the best possible way to deliver the relationship between composition and its resulting biological activity. In our interpretation, this appears to us to be very correctly also the spirit of the reviewer’s concern. Building on the know-how we accumulated over the R&D process, we concluded that it would be correct, and were comforted by the agreement expressed by regulatory authorities, to refer to the contribution of polysaccharides and resins to the adhesion and barrier properties of the product. It is of pivotal important, though, to consider that for instance the adhesive and barrier properties of the final product cannot be reconducted exclusively to the presence of these single fractions but to properties that emerge contemporarily in the context of the final formulation and are determined by the interactions between the different components. As a consequence of this we actually also centered the batch release policy of the product on reproducibility of these parameters, which are the ones underlying its efficacy. We sincerely hope that in the light of this explanation, the reviewer will be able to accept our revision of the manuscript, in which we add an additional patent and quantitative information with respect to the abundance of different extracts in the final product.

- Question 3. Conclusions: author claim that: “Our data confirm the ability of Colilen IBS® to protect the intestinal mucosa against irritating and inflammatory agents…” however several lines above in the discussion section author admit that “Although pre-treatment with Colilen IBS® does not prevent the development of DNBS-induced damage…”. Thus, this overstatement has to be removed from the conclusions.

Response:  From our results it emerges that the effect of Colilen IBS® in protecting the intestinal mucosa from the damage is time-dependent. In fact, in animals pre-treated with Colilen IBS®, 3 days after administration, the tissue appears damaged, while at later times, a progressive healing of the intestinal mucosa is observed, with an overall protection effect. We agree with reviewer that the expression “Our data confirm the ability of Colilen IBS® to protect the intestinal mucosa against irritating and inflammatory agents….”  is not entirely appropriate and, for this reason, we modified it in " Our data confirm the ability of Colilen IBS® to protect the intestinal mucosa and to promote its healing from lesions, as demonstrated at histological level. ", in conclusions (lines 452-453).

- Question 4. Minor issues: Abstract: „Inflammatory Bowel Syndrome” should be corrected since the correct name id irritable bowel syndrome.

Response: We apologized with the reviewer for this error. We have corrected the name (line 14).

- Question 5. Minor issues. Abstract and discussion: “device” is not a proper word to use when describing a plant-derived product. Please amend.

Response: We would like to address the concerns of Reviewer #1 with respect to the definition of Colilen IBS® as a device in the “abstract and conclusion” section. Unfortunately, the correct terminology regarding products containing plant derivatives is quite confuse and not homogenous in different countries. Therefore, we have to refer to the legislation under which Colilen IBS® commercialization is authorized, that is, EU Directive 93/42/EC (as amended), the so called “Medical Device Directive”. A variety of insights and implications deriving from the application of the Directive are reviewed in the publication attached as Annex I to this document. Such a Directive provides for a legislative environment in which efficacy, safety and production quality of therapeutic solutions are thoroughly reviewed and validated by certified third parties, in order to comply with extremely high standards than can protect patients from opportunistic behaviors potentially exhibited by manufacturers. We believe that the “Medical Device Directive” represents a strategic opportunity in order to make innovative therapeutic solutions available to patients, especially when of plant origin. This is a fact, actually already reshaping the medicinal products market by enabling innovative, efficacious and safe therapeutic solutions to efficiently come to the market (worth almost 1 billion euros in Italy alone, in 2018). Further confirmation of the general acknowledgement of the strategic importance of this sector derives from the recent confirmation of the “Medical Device Regulation” through the approval of Regulation 745/2017, in which a specific rule (Rule 21) is dedicated exclusively to the classification of Medical Devices Made of Substances. As broadly acknowledged, in fact, complex mixtures of compounds of natural origin quite simply cannot comply with drug legislations designed to deal with isolated, possibly synthetic single molecules. In contrast with what is technically achievable for single molecules, basic requirements of any drug-norming legislation (affinity, recognition site, residence time, selectivity, distribution) can only be predicted but factually not experimentally demonstrated for molecules within a complex mixture of natural origin. Knowledge of these aspects currently do not represent a requirement of the Medical Device Directive and, when stemming from an R&D pipeline ad hoc designed in order to accommodate for specific demands in terms of mechanism of action, the resulting complex mixtures of natural origin can indeed represent a credible and innovative therapeutic tool available for both patients and medical doctors. We sincerely believe that, despite its peculiarities, this represents a path to drag therapeutic options deriving from natural sources from the area of alternative medicine to that of allopathic medicine, a very much needed process in order for mankind not to lose a plethora of options derived from nature. Attached as Annex II to this document we hereby enclose the confidential CE certificate confirming the compliance of the technical dossier with the applicable requirements of Directive 93/42/EC, and is the reason why we referred to the product as a device and ask to be allowed to keep doing so. In this revision of the manuscript we tried to clarify this aspect by introducing limited changes to the text in order to comply with reviewers prescriptions and correct misprint which led to rephrasing in order to streamline a sentence. We also sincerely hope that the views on such matters and the categorization of the product, will be shared by Nutrients and the reviewers dedicated to reviewing our work, at least to an extend capable of granting permission for publication on the journal of our manuscript in this revised version.

- Question 6. Minor issues. Discussion: DNBS model should not be referred to as IBS model since it induces colonic inflammation which may or may not cause IBS. DNBS should be rather referred to as model of colitis.

Response: As describe Adam et al. 2006 and Qin et al. 2012 (Adam, B.; Liebregts, T.; Bertram, S.; Holtmann, G. Functional gastrointestinal disorders. Dtsch Med Wochenschr. 2006;131(45):2531-40; Qin, X. Etiology of inflammatory bowel disease: a unified hypothesis. World J Gastroenterol. 2012; 1708–1722), DNBS was frequently used as model of post-inflammatory IBS, in fact the animals treated with DNBS keep showing visceral hypersensitivity and intestinal dysmotility also after the resolution of inflammation and intestinal damage. Anyway, we accept the correct suggestion of the referee and we change the expressione "DNBS was frequently used as model of IBS" in "DNBS was frequently used as model of colitis" (on line 358).

Reviewer 2 Report

I read with interest the MS" Research of new therapeutic approaches for  abdominal visceral pain treatment: preclinical Colilen IBS® effects by Parisio C and coworkers". It is a well designed MS dealing with a demanding issue and I understand the enthusiasm of the Authors for their results. However, the Authors should highlight the preliminary nature of the study both on the abstract and throughout the whole MS: it is  small sample study on rats and results cannot be generalized to such a complex scenario as IBS-IBD are in human beings. I would not dictate how the MS should be improved, but statements as "We suggest that Colilen IBS® represents a new and valid therapeutic strategy to manage IBS and IBD" are simply not consistent  with the study results. In addition, I suggest to be more precise  when describing drug actions: steroid display a complex Tx mechanism when used to treat IBD, but the so-called cell protective effort is likely a minor one at best Finally, A clearer description of the methods used to perform the study would be very much welcomed by the non-focused reader.

Author Response

Comments and Suggestions for Authors. I read with interest the MS "Research of new therapeutic approaches for abdominal visceral pain treatment: preclinical Colilen IBS® effects by Parisio C and coworkers". It is a well-designed MS dealing with a demanding issue and I understand the enthusiasm of the Authors for their results. However, the Authors should highlight the preliminary nature of the study both on the abstract and throughout the whole MS: it is small sample study on rats and results cannot be generalized to such a complex scenario as IBS-IBD are in human beings. I would not dictate how the MS should be improved, but statements as "We suggest that Colilen IBS® represents a new and valid therapeutic strategy to manage IBS and IBD" are simply not consistent with the study results. In addition, I suggest to be more precise when describing drug actions: steroid display a complex Tx mechanism when used to treat IBD, but the so-called cell protective effort is likely a minor one at best. Finally, a clearer description of the methods used to perform the study would be very much welcomed by the non-focused reader.

Response: We are pleased for the interest of the reviewer for our MS. We agree with him that this is a preclinical evaluation of the effect of Colilen IBS® on the intestinal damage and visceral pain development in the rat model of colitis. We are conscious that the management of visceral pain is a serious clinical problem in patients with IBD-IBS and that the scenario in these pathologies is very complex. Furthermore, the therapeutic treatment for chronic visceral pain is still unsatisfactory and, although the knowledge in these pathologies is progressing, new therapeutic strategies are needed. Therefore, as suggested by the reviewer, we changed the expression "We suggest that Colilen IBS® represents a new and valid therapeutic strategy to manage IBS and IBD" in "We therefore suggest that Colilen IBS® could represent an innovative therapeutic option to pain management in bowel diseases" in conclusios (lines 455-456).

Round 2

Reviewer 1 Report

The manuscript has been improved. I find changes made by the authors satisfactory. However I could not find author's rebuttal to one of my major concerns raised in the previous review, namely:

Figure 1. The VMR to CRD in control animals (vehicle + vehicle) does not change with the volume of the balloon. How do authors explain it? Even in control animals a response to increasing volume of balloon should be clearly visible. Otherwise it may be the case that the signal is not coming through electrodes to the amplifier or there are some other issues related to signal transduction in this setup. Furthermore, as showed on Fig. 2 VMR to CRD increases in control animals (approx. 100% increase) as the pressure increases.

Please address this comment.

Author Response

As reported by Christianson and Gebhart (Christianson and Gebhart. Assessment of colon sensitivity by luminal distension in mice. Nat. Protoc. 2007, 2(10):2624-31), the Colon-Rectal Distension (CRD) caused by balloon swelling is a mechanical stimulus that replicates the sensation and referral pattern of visceral pain. For this reason, in all our experiments the Viscero-Motor Response (VMR) to CRD was used as objective measure of visceral sensitivity in the animals.

Before to start with the experiments, the goodness of the electrodes implanted in the abdominal muscle of all animals was evaluated: this was possible because, damaged or incorrectly positioned electrodes give a distorted electromyographic signal, which is very different from that coming from intact electrodes and properly functioning. Animals for which a correct EMG trace was not observed, were excluded from the experiments. In addition, during the experimental procedures, the animals are continuously monitored by the operator, who observes any abdominal muscle contraction during the CRD. In the case of control animals, during the swelling balloon no abdominal muscle contraction was observed, confirming the recorded EMG signal. We confirm that in our animal model, the control animals (vehicle + vehicle) show a very low VMR to increasing volume of balloon in control condition. On the other hand, this response seems to be comparable to results obtained by other authors in previous papers, where no significant differences were reported in response to increasing volumes or pressures, please see:

- Shah MK, Wan J, Janyaro H, Tahir AH, Cui L and Ding M-X (2016) Visceral Hypersensitivity Is Provoked by 2,4,6-Trinitrobenzene Sulfonic Acid-Induced Ileitis in Rats. Front. Pharmacol. 7: 214

Furthermore, it is important to underline that in our experiments, the measurements of VMR to CRD are carried out in animals under light anesthesia (2% isoflurane) in order to allow a correct and stable EMG registration, and this treatment reduces the basal sensitivity of animals. In our opinion this is a very relevant difference that could explain the reason why other reports describe an increasing response in basal condition; for your convenience a list of article that do not use anesthesia are as follows:

- Nadine El-Ayache1 and James J. Galligan. 5-HT3 receptor signaling in serotonin transporter-knockout rats: a female sex-specific animal model of visceral hypersensitivity. Am J Physiol Gastrointest Liver Physiol 316: G132–G143, 2019;

- Nakaya K, Nagura Y, Hasegawa R, Ito H, Fukudo S. Dai-Kenchu-To, a Herbal Medicine, Attenuates Colorectal Distention-induced Visceromotor Responses in Rats. J Neurogastroenterol Motil. 2016;22(4):686–693.

We sincerely hope that in the light of this explanation, the reviewer will be able to accept our revision of the manuscript.

Reviewer 2 Report

I read with interest the revised version of this MS dealing with alternative TX of bowel disorders. All of my queries have been addressed in full. No additional suggestions on this side.

Author Response

We are grateful to the reviewer for the interest and attention with which he has reviewed our MS.